# Clothing the Emperor: Dynamic Root–Shoot Allocation Trajectories in Relation to Whole-Plant Growth Rate and in Response to Temperature

**DOI:** 10.3390/plants8070212

**Published:** 2019-07-10

**Authors:** David Robinson, John Henry Peterkin

**Affiliations:** 1School of Biological Sciences, University of Aberdeen, Aberdeen AB24 3UU, UK; 2School of Biological Sciences, University of Portsmouth, Portsmouth PO1 2DY, UK

**Keywords:** allocation, biomass, forb, grass, relative growth rate, root mass fraction, root–shoot, temperature

## Abstract

We quantified how root–shoot biomass allocation and whole-plant growth rate co-varied ontogenetically in contrasting species in response to cooling. Seven grass and four forb species were grown for 56 days in hydroponics. Growth was measured repeatedly before and after day/night temperatures were reduced at 28 days from 20 °C/15 °C to 10 °C/5 °C; controls remained unchanged. Sigmoid trajectories of root and shoot growth were reconstructed from the experimental data to derive continuous whole-plant relative growth rates (RGRs) and root mass fractions (RMFs). Root mass fractions in cooled plants generally increased, but this originated from unexpected and previously uncharacterised differences in response among species. Root mass fraction and RGR co-trajectories were idiosyncratic in controls and cooled plants. The RGR–RMF co-trajectories responded to cooling in grasses, but not forbs. The RMF responses of stress-tolerant grasses were predictably weak but projected to eventually out-respond faster-growing species. Sigmoid growth constrains biomass allocation. Only when neither root nor shoot biomass is near-maximal can biomass allocation respond to environmental drivers. Near maximum size, plants cannot adjust RMF, which then reflects net above- and belowground productivities. Ontogenetic biomass allocations are not equivalent to those based on interspecific surveys, especially in mature vegetation. This reinforces the importance of measuring temporal growth dynamics, and not relying on “snapshot” comparisons to infer the functional significance of root–shoot allocation.

## 1. Introduction

At its most basic, the question of how a terrestrial vascular plant allocates its biomass addresses variations in the distribution of growth between aboveground parts (i.e., shoots, comprising stems, leaves and flowers) and those belowground (roots) among species, over time, in response to environmental conditions and in relation to productivity. This root–shoot dichotomy, as simplistic as it is, is fundamental to a plant’s existence because roots and shoots do different but equally essential and complementary things. More roots can capture more nutrients and water, but more leaves can intercept more light and fix more carbon dioxide. Plants need all these resources to produce biomass, compete with neighbours and reproduce, and they require both leaves and roots to operate in a coordinated way to meet that need.

Ontogenetic adjustments in root–shoot biomass allocation are thought to help a plant achieve some compensation for resource scarcity, physical damage or harsh conditions to which the plant may be subjected during its life [1,2]. “Compensation” in this context is generally understood to mean that some measure of growth, survival or fitness is improved by temporary or permanent changes in allocation beyond that which would be possible in the absence of that response [3]. Embedded in this is the idea that allocation can, in theory, be optimal (or not) with respect to growth, survival or fitness. This principle has inspired much modelling activity. For all these reasons, variations in root–shoot biomass have been measured, catalogued and modelled for decades [4,5,6,7].

Despite this trove, root–shoot biomass allocation and its relationships with whole-plant growth rate and environmental conditions continue to resist complete understanding. Among the reasons for this are: confusing the temporal changes in allocation with size-dependent differences in allocation [5]; confusing ontogenetic changes in root–shoot biomass of individuals with interspecific variations in root–shoot biomass among plants differing in size [8]; inaccurately measuring root mass [9]; the difficulty of attributing changes in allocation as genuine responses to environmental conditions rather than to “ontogenetic drift” [10] (p. 16); the limited information provided by measurements made, as they usually are, on only one or a few occasions, or on only one or a few species, so that the temporal dynamics and phylogenetic breadth of allocation responses are poorly resolved [3]; the difficulties of relating instantaneous measurements of allocation to non-instantaneous estimates of growth rate [11]; the lack of a simple framework to reconstruct allocation trajectories and their co-variations with growth rate in response to environmental factors, with which limited experimental data can then be compared [12,13]. The resulting illusory understanding of root–shoot allocation led one critic to invoke the memorable and revealing metaphor of the Emperor’s New Clothes [5].

Here, we address these issues within a single experiment. We compared growth and root–shoot biomass allocation in contrasting herbaceous species to test if these interspecific differences produced measurable ontogenetic differences in allocation before and after a defined environmental change guaranteed to influence root or shoot growth rates—a reduction in ambient temperature. Plants were grown hydroponically to maximise the accuracy of root biomass measurements, and in controlled environments to minimise unknown external influences on growth and allocation. Root and shoot growth were measured repeatedly by destructive sampling to allow temporal changes in allocation to be detected.

Our analysis was based on the following simple propositions:Root and shoot growth each has a sigmoid trajectory with time. Growth is initially slow, followed by a linear or exponential phase of faster growth. Growth rate falls gradually, eventually approaching zero when there is no further net increase in biomass, typical of annuals, but many species, including perennials, grow like this for at least part of their lives [14,15]. Biomass increase is halted by factors including self-shading, inter-root competition, resource depletion, crowding, tissue turnover, transitioning from vegetative to reproductive growth, determinate development, dormancy, photoperiodic downregulation of metabolism, senescence and numerous environmental and biotic constraints, depending on species and circumstances. An ontogenetic response of root–shoot allocation to the environment is defined as a deviation in allocation following a change in environmental conditions compared with allocation measured in control plants. In most root–shoot allocation studies, plants are subjected to different but temporally static environmental conditions. Often, no temporal information about allocation is collected, so it is not always clear what constitutes ontogenetic drift in allocation as distinct from a genuine response. But if some plants are subjected to a specific treatment at a defined time, and if growth is measured repeatedly before and after that change, and compared with controls, it is possible to say definitively if allocation responds to that treatment; temporal changes in allocation in control plants then reflect ontogenetic drift [16,17,18,19,20,21,22]. A biomass allocation response to the environment can occur only if there is a differential change in rates of biomass production between root and shoot [1]. That is not to say there can be no response at all without biomass change, but it will be confined to adjustments in physiological processes such as specific rates of photosynthesis, respiration, water uptake, nutrient capture and so on, which do not necessarily involve the production of new biomass; such processes are obviously important but are not considered here [23,24].

We used experimental data [25] to reconstruct separate statistically modelled trajectories of root and shoot biomass as continuous functions of time. From these, we derived instantaneous estimates of whole-plant growth rate and root–shoot biomass allocation to show how these variables co-vary over time and in relation to whole-plant size. Combining empirical information with statistical modelling of biological systems, whether individuals, populations or communities, is a powerful way to better understand their dynamics. Such approaches have a long history [10,14,26] and continue to provide new insights [13,27].

Although we used a temperature change to experimentally modify root and shoot growth rates, our aim was not to test hypotheses about plants’ thermal physiologies. Of course, temperature has profound influences on growth and allocation during ontogeny [23] and across ecosystems [28]. Our primary aim was to clarify and compare how root–shoot biomass allocation by individual plants might vary with species, time, size, growth rate and in response to cooling, and how allocation could be constrained by a plant’s growth trajectory; the operative words here are “might” and “could”.

## 2. Materials and Methods 

### 2.1. Experimental

Eleven herbaceous species were chosen [25] to represent a range of life-histories, ecologies, phenologies, growth rates and morphologies according to criteria based on comparative data [29] (Appendix A): seven grasses (*Anthoxanthum odoratum* L., *Arrhenatherum elatius* (L.) J. & C. Presl, *Catapodium rigidum* (L.) C.E. Hubb., *Deschampsia flexuosa* (L.) Trin., *Festuca ovina* L., *Holcus lanatus* L. and *Poa annua* L.) and four forbs (*Cardamine hirsuta* L., *Centaurea nigra* L., *Rumex acetosa* L. and *Scabiosa columbaria* L.); nomenclature follows Reference [30].

Plants were grown hydroponically in a controlled environment [31]. Seeds were germinated on 0.1 strength Rorison solution with nitrate as the sole nitrogen source at a concentration of 0.4 mmol L^−1^ [32]. Seedlings were transplanted 14 d after emergence into 1 L containers supplied with a continuous flow of aerated nutrient solution at 0.3 L h^−1^ delivered from 20 L reservoirs via peristaltic pumps. Solution pH in the containers was measured daily. A pH of 4.5 (±0.2) was maintained by adding between 0.1 and 5 mL of 0.5 M sulphuric acid to each reservoir.

Controls comprised five replicate vessels per species, each initially containing 14 seedlings. Vessels were arranged in a fully randomised design and maintained for 56 d at air temperatures of 20 °C day, 15 °C night, 70% relative humidity, 16 h daylength and irradiance 50 W m^−2^ provided from a combination of fluorescent tubes and tungsten bulbs. Plants were harvested after 7 d then at weekly intervals up to 56 d, eight harvests in total. The first, second and third harvests consisted of four, three and two plants per replicate, respectively, to provide material for chemical analyses for a separate study (data not shown). Harvests four to eight were of one plant per replicate.

An adjacent identical growth room was used to cool treated plants. The setup was as for the control for the first 28 d, when, over a period of four hours in the middle of the photoperiod, the temperature was reduced gradually to 10 °C and then maintained at 10 °C day, 5 °C night. Simultaneously, relative humidity was lowered to 62% to minimize changes in saturation vapour pressure deficit associated with cooling. Treatment harvests five to eight each comprised one plant per replicate, as in the controls.

Plants were selected for harvesting by initial, random assignment of each seedling to a harvest. Harvested plants were separated into roots and shoots, oven-dried for 48 h at 80 °C and dry weights determined.

### 2.2. Data Analysis

The data available for analysis [25] were the mean whole-plant dry weights (g per plant) at each harvest calculated according to Reference [33], along with corresponding root:shoot ratios. From these, mean root and shoot biomasses were calculated (Appendix A). 

Incremental changes in root or shoot biomass between two successive times, *t*_1_ and *t*_2_, and which form part of the assumed sigmoid trajectory (see Introduction), can be described by a difference equation such as the two-parameter logistic familiar from population biology:*Y_t_*_2_ = *Y_t_*_1_ (*t*_2_ − *t*_1_) (1 + *r*[1 − (*Y_t_*_1_/*Y_max_*)])(1)
where *Y* is root (*R*) or shoot (*S*) biomass (dry weight; g per plant), Y_max_ the maximum (asymptotic) value of *Y*, *t* time (d), and *r* the intrinsic growth rate of *Y* (d^−1^). The absolute growth rate of *Y* at any point is *rY*(1 − (*Y*/*Y_max_*)). Equation (1) is one of many mathematical descriptions of sigmoid growth that could be used for this analysis [14], but probably the simplest.

Equation (1) was fitted to weekly mean biomass data by simultaneously adjusting *r* and *Y_max_* to maximise goodness-of-fit between data and model [13]. This was done separately for the roots and shoots of each species. Goodness-of-fit (*R*^2^) exceeded 0.959 in all cases (Appendix A). For cooled plants, data were identical to those in controls up to 28 d when the temperature was reduced. 

Values of *r* and *Y_max_* derived by fitting Equation (1) to the data (Appendix A) were used to generate temporal trajectories of root and shoot biomasses at daily resolution from the first harvest at 7 d to the last at 56 d. For controls, the same species-specific *r* and *Y_max_* values were used throughout. For the cooled plants, those *r* and *Y_max_* values were used up to 28 d because their growing conditions had been identical to the controls until that time. Thereafter, the *r* and *Y_max_* values derived for cooled plants were used. This assumed a sudden rather than gradual adjustment in growth rates, which introduced small discontinuities into the resulting trajectories of cooled plants around 28 d.

The derived trajectories were also extrapolated to explore potential changes in biomass allocation beyond 56 d, to 120 d, by which time the species used in the experiment would probably have approached steady-state growth. Normally forbidden, such extrapolations are justified here because: (a) it is likely that growth was sigmoid for the reasons given earlier (it would have been less appropriate to assume linear or exponential extrapolations, for example); (b) Equation (1) is mathematically symmetrical, i.e., if the lower part is defined by a strong fit to data (as it was) then the upper extrapolated portion is defined automatically, which of course assumes no late-acting influences on the trajectory or developmental discontinuities such as a switch from vegetative growth to flowering; (c) we did not expect the extrapolations to predict precisely a plant’s ultimate size, but merely hint at a possible trajectory towards that point; (d) this approach can inform subsequent experiments designed to test if real allocations match those suggested by the extrapolations and to identify possibilities that are “worthy of additional investigation or scepticism” [34]; (e) extrapolated allocations were checked to ensure they were realistic (see below); and (f) we clearly distinguished extrapolated trajectories from those coincident with data.

Root–shoot biomass allocation was expressed as the root mass fraction (RMF; [4]):RMF = *R*/(*R* + *S*)(2)

The RMF can range, in theory, from 0 to 1, although the practical range of variation is narrower than this for physiological and physical reasons. Daily values of *R* and *S* generated by Equation (1), as fitted to data, were used to derive temporal trajectories of RMF using Equation (2). A response of root–shoot allocation to cooling at 28 d is defined here as a subsequent difference in RMF trajectories between cooled and control plants. Temporal changes in RMF of controls are defined as ontogenetic drift. 

Whole-plant growth rate is usually expressed as relative growth rate (RGR). Mean RGR over a time interval *t*_1_ to *t*_2_ is calculated in terms of root and shoot biomasses as:RGR = [ln (*R_t_*_2_ + *S_t_*_2_) − ln (*R_t_*_1_ + *S_t_*_1_)]/(*t*_2_ − *t*_1_)(3)

If the interval *t*_2_–*t*_1_ is sufficiently short compared with the entire growth period, Equation (3) approaches a continuous estimate of RGR [14] (p. 18). Again, daily values of *R* and *S* generated by Equation (1) were used to derive near-instantaneous trajectories of RGR from Equation (3). This overcomes the valid objection [11] that it is meaningless to compare an instantaneous biomass fraction (RMF) with a biomass flux (RGR) estimated non-instantaneously. The continuous trajectories of RMF from Equation (2) were plotted against corresponding RGR trajectories from Equation (3) to visualise their co-variation [13]. Note that, in terms of Equation (1), RGR is related instantaneously to *r* since RGR = *r*(1 − (*Y*/*Y_max_*)).

Temporal trajectories of RMF and RGR were also plotted as functions of whole-plant biomass (i.e., *R* + *S*) to detect size dependencies.

Allometric analysis is often considered the “gold standard” with which to quantify co-variation between root and shoot growth across the whole period of interest. Root biomass is assumed to vary as a power function of shoot biomass [35] (p. 15):*R* = β*S*^α^(4)
where α is a scaling exponent and β a scaling coefficient. When Equation (4) is log-transformed, α becomes the slope of the linear regression between ln R and ln S and ln β its y-intercept:ln *R* = ln β + α ln *S*(5)

In this guise, α is the quotient of root and shoot RGRs [11]. When α = 1, root and shoot biomasses vary isometrically; when α < 1, shoots grow faster than roots and the plant grows progressively more “shooty”; when α > 1, more “rooty”. β has no distinct biological meaning [11]. Comparisons of α and β in different environments can be informative, especially in helping to distinguish genuine root–shoot responses from those that are simply correlates of plant size [8,11].

α and β values were derived from control and treatment data for each species as the slopes and intercepts, respectively, of reduced major axis regressions [35] (p. 328) of ln *R* on ln *S* (Equation (5)) using, first, only the experimental data from 7–56 d (*n* = 8) and, second, the extrapolated root and shoot biomasses derived from Equation (1) from 7–120 d as daily values (*n* = 114). The latter was done to check if the root–shoot allometries derived from extrapolated trajectories were plausible. From root and shoot dry weights reported for many herbaceous angiosperm species grown in a range of light and nutrient environments [36,37,38], mean α = 1.02 (*n* = 224) with values ranging from 0.47–1.90. Values of α falling in this range can therefore be regarded as realistic.

## 3. Results

### 3.1. Root and Shoot Growth

The trajectories of growth and biomass allocation in Figure 1 and Figure 2 have some surprising similarities and differences. Among the not-so-surprising findings is that cooling at 28 d slowed the growth of all species, but some more so than others. Whole-plant RGRs of the slow-growing species of *Catapodium*, *Deschampsia* and *Festuca* were barely affected by cooling compared with the controls; those of *Arrhenatherum*, *Poa*, *Rumex* and *Scabiosa* declined dramatically.

Cooling reduced shoot growth in all species, but not necessarily root growth. For example, shoot biomass of cooled *Deschampsia*, *Festuca* and *Holcus* at 56 d was between one- and two-thirds of the controls, but root growth was unaffected, trends amplified by the extrapolated trajectories derived using Equation (1). None of the species had likely reached their ultimate sizes by 56 d; Equation (1) suggested that all, apart from *Deschampsia* and *Festuca*, could have done so by 120 d. There was a suggestion from the extrapolated trajectories that in *Catapodium*, *Deschampsia* and *Festuca*, RGRs of control and cooled plants might have crossed over after about 70 d, with cooled plants eventually growing faster than controls (Figure 1j,n,r).

All plants remained vegetative throughout the experiment. Therefore, none of the measured variations in biomass allocation up to 56 d were caused by a shift from vegetative to reproductive growth.

Fitting Equation (1) to the root and shoot biomass data of cooled plants required, in almost all cases, a large reduction in *Y_max_* compared with controls (Appendix A). The exceptions were for root biomass in *Catapodium*, *Deschampsia*, *Festuca* and *Holcus*, for which the data suggested increased *Y_max_* for cooled plants. In no case was it necessary to substantially change values of *r* for cooled plants from those of controls.

### 3.2. Root and Shoot Biomass Allocation 

Up to 56 d, RMF remained near-constant (*Arrhenatherum*, *Catapodium*, *Rumex*), declined gradually (*Anthoxanthum*) or increased gradually (*Centaurea*) in the controls, reflecting different ontogenetic drifts in allocation (Figure 1). Beyond 56 d, RMF might have increased in the controls having previously declined (*Deschampsia*, *Scabiosa*), whereas in others, the downward drift possibly continued (*Holcus*, *Poa*). We note, however, the likelihood that *Poa*, an annual species (Appendix A), would probably have flowered at some point before 120 d, with potential effects on root–shoot biomass allocation not reflected by the trajectories shown in Figure 1y–bb.

In response to cooling at 28 d, RMF always increased by 56 d except in *Rumex*, which did not respond (Figure 1 and Figure 2). The RMF responses varied among species. By 56 d, RMF trajectories in response to cooling had diverged substantially from controls in *Anthoxanthum* and *Holcus*, more modestly in *Arrhenatherum*, *Poa* and *Centaurea*, and scarcely in *Catapodium*, *Deschampsia* and *Festuca*. After 56 d, however, those insubstantial responses of RMF in the slower-growing species could have developed into some of the strongest, particularly in *Catapodium* and *Festuca*. 

Plotting the RMF and RGR trajectories as functions of whole-plant biomass reveals more similarities than differences among species in response to cooling (Figure 1 and Figure 2). The RGR always declined with increasing plant size, predictable mathematically from Equation (1), and at the lower temperature, always declined more steeply with size compared with controls. At the lower temperature, RMF always increased more steeply with size than in controls, except in the unresponsive *Rumex*. 

The plausibility of the extrapolations of Equation (1) between 56–120 d is supported by comparing allometric analyses based only on the experimental data (7–56 d) with those based on the derived trajectories (7–120 d). Allometric coefficients (α) based on data alone ranged from 0.78–1.20 in controls, and 0.88–1.50 in cooled plants (Appendix A). The corresponding α values derived from extrapolated trajectories were 0.84–1.20 and 1.06–1.50, so falling comfortably within the realistic range of 0.47–1.90 (see Section 2.2). 

### 3.3. Co-Variation between Allocation and Whole-Plant Growth Rate

The co-variations between RMF and RGR were among our most intriguing results. Those for *Anthoxanthum* (Figure 1d) illustrate some of the main features. In controls, up to 56 d, RMF and RGR were strongly and positively correlated (Appendix A). That correlation was reversed in cooled plants in which RMF and RGR were negatively correlated. When the possible influence of the extrapolated trajectories was included, there was no correlation between RMF and RGR up to 120 d in the controls. At the lower temperature, the negative correlation was strengthened. Most of the other grasses behaved similarly (Figure 1d,h,l,p,t,x,bb), although extrapolated trajectories of *Deschampsia* and *Festuca* were distinctly more circuitous between 56–120 d. The RMF responses in *Deschampsia* and *Festuca* by 56 d were modest, but projected to become far larger by 120 d. Overall, there was little evidence of a consistent relationship between allocation and whole-plant growth rates across the seven grass species.

By contrast, RMF and RGR always co-varied closely in the forbs (Figure 2d,h,l,p). Cooling had relatively little effect on how RMF and RGR co-varied. Trajectories of control and cooled plants virtually overlapped.

## 4. Discussion

### 4.1. Response of Root–Shoot Allocation to Cooling 

A meta-analysis of how root–shoot allocation varies with temperature [6] concluded that RMF generally increases in cooled plants and, indeed, cooling increased RMF in 10 of our 11 species, the exception being *Rumex*. However, our analysis revealed that that response of RMF originated from interspecific differences in how root and shoot growth rates responded to cooling. That conclusion could not have been reached without reconstructing, via Equation (1), the separate and detailed growth dynamics of roots and shoots before and after cooling.

### 4.2. Co-Variation between Allocation and Whole-Plant Growth Rate

We found a striking difference between grasses and forbs in how closely root–shoot allocation co-varied with whole-plant growth rate. In forbs, the co-variation between RMF and RGR followed essentially the same trajectories irrespective of temperature, suggesting an inflexible coupling between allocation and production of biomass in those species. This was true for growth up to 56 d and for trajectories extrapolated to 120 d. The relative inflexibility in the RMF–RGR responses in forbs could be a function of the cooling regime imposed in this experiment rather than it being a fixed trait. Forbs and other dicots can exhibit large root–shoot allocation responses to environmental variability [21,39,40,41]. 

The RMF–RGR co-trajectories in grasses were less strongly coupled than in forbs. Cooling produced a larger RMF response in grasses for a given change in RGR. The size of that response was species-dependent. There were large responses in *Holcus* by 56 d in comparison with those in *Deschampsia* and *Festuca*, which resembled those of the unresponsive forbs. However, the RMF–RGR co-trajectories of *Deschampsia* and *Festuca* were projected to diverge more widely than for any species by 120 d, although the uncertainty associated with the extrapolations is, of course, acknowledged. Nevertheless, this is a testable possibility. 

The RMF–RGR co-trajectories plotted in Figure 1 and Figure 2 represent a fraction of the potential co-variations in these traits. Some RMF–RGR combinations are untenable: a fast RGR obviously cannot be attained by any species if its RMF approaches 0 or 1. The outer envelope of biophysically possible combinations of RMF and RGR is currently impossible to determine; applications of metabolic scaling theory might be useful here [42]. 

Unfortunately, the interspecific idiosyncrasies of the RMF–RGR trajectories plotted in Figure 1 and Figure 2 offer little encouragement for a theoretician hoping to develop models linking biomass production and allocation. It is unlikely that a simple mechanistic model of root–shoot allocation would have predicted those trajectories, although a structurally complex and parameter-rich model [43] could have done so. Even now, much remains to be done simply to characterise experimentally the full spectrum of co-variation between root–shoot allocation and whole-plant growth in response to different environmental factors (including rhizosphere microbes and neighbours [44,45]) and among a wider range of species [3]. Likewise, exploring the wider dynamical behaviour of Equation (1) or its analogues as a simple mathematical description linking root–shoot allocation with whole-plant growth could yield valuable insights into, for example, how allocation might be constrained from providing a greater compensatory effect, should such an effect be possible. 

### 4.3. Compensating for Something?

Is there any evidence in Figure 1 and Figure 2 for compensatory adjustments in root–shoot allocation in response to cooling, or for the attainment of a new optimal growth rate or state by cooled plants? The RGR in cooled plants never exceeded the corresponding RGR in controls irrespective of how RMF responded to cooling up to 56 d. However, some RGR trajectories extrapolated beyond that time did cross over, those of cooled plants eventually exceeding those of controls, albeit at very slow rates of growth in already inherently slow-growing species (*Catapodium*, *Deschampsia*, *Festuca*). Consequently, the potential compensatory effect of any allocation response could have been only marginal at best. The possibility of RGRs of cooled and control plants eventually crossing over in some species merits further investigation.

Other than the trivial findings that as RGRs declined progressively from initially maximum rates, as dictated by the mathematics of sigmoid growth (Equation (1)), RMFs also declined, and that if RMF approached its upper or lower limit, RGRs would inevitably have reached zero (Figure 1 and Figure 2), we found no convincing evidence that a faster RGR was somehow enabled by plants adjusting their RMFs optimally in response to cooling. When experiments have been designed with the explicit aim of testing the idea of optimal root–shoot allocation, plants track theoretically optimal trajectories only coincidentally, if at all [46,47]. Other studies have found little or no evidence for optimal allocations in response to environmental variability [21,48,49], although some do make this claim [50]. 

Rather than invoking optimality to explain observed (and extrapolated) allocation responses to cooling, a more parsimonious interpretation is that RMF responses in cooled plants simply reflect unavoidable temperature-induced changes in relative root and shoot growth rates. If correct, that interpretation says that changes in root–shoot allocation in response to cooling are not primarily adaptive. That same point emerged from a study [20] that found no evidence for adaptive changes in root–shoot allocation in response to switches in nutrient supply despite whole-plant growth rate responding to that treatment. At least with reference to temperature, it might be time to acknowledge that it can be unhelpful to assume that plants, with their multiple and dispersed growing points, mimic precision-engineered machines in coordinating their biomass allocation responses to external cues in a deterministically optimal way, despite the evident structural and physiological coordination between roots and shoots. Whether the same conclusion applies generally to plants responding to limited resources such as nutrients, water or light remains to be seen.

### 4.4. Growing Fast or Slow

Plant strategy theory says that stress-tolerant species respond to unfavourable environments with weak phenotypic changes compared with those of faster-growing species [51] (p. 91). Stress-tolerant plants tend instead to have evolved traits associated with resource conservation, herbivore and pathogen deterrence to favour long-term survival of vegetative meristems and other structures. Many experiments support the view that stress-tolerant species are less responsive morphologically than their more vigorous counterparts [52], as do the RMF responses in slow-growing *Festuca* compared with *Holcus*, for example, measured by 56 d (Figure 1q,v). 

However, extrapolating allocation beyond 56 d suggested that *Festuca*’s RMF response to cooling eventually could have exceeded that of *Holcus*. The predictably weak RMF response of *Festuca* to cooling between 28–56 d was perhaps just a slow response. This suggestion reinforces the importance of having a temporal perspective on plants’ responses to their environment. It matters when in the plants’ growth trajectories and life-histories traits comprising phenotypic responses are measured. Experiments, surveys or screening programmes in which traits are measured only once or a few times can provide incomplete or even misleading information about the nature of those traits, despite the greater effort needed to quantify them repeatedly over long periods [3]. 

Ecologically it probably matters little if a species’ weak responses are just slow responses compared with those of a competitor. Over the short-term, the species that responds most rapidly will have the larger response and probably out-compete its slower neighbour in terms of resource capture [13,53]. But at longer timescales, other traits influence competitive dominance [51] (p. 179–198). In the case of *Festuca*, it sustains continued biomass production above- and belowground over winter, whereas other perennials with which it co-exists, such as *Arrhenatherum*, lose biomass through leaf and root senescence in autumn and winter [54]. Across seasons, subtle adjustments in RMF will be subsumed into biomass gains and losses influenced by depredations of temperature extremes, herbivory, disease and decomposition.

### 4.5. Experimental Designs and Analytical Approaches

Is a purely descriptive model of plant growth useful? Arguments for and against fitting phenomenological models such as Equation (1) rather than polynomial or spline functions to plant growth data have been aired comprehensively [14,26], as have those for mechanistic models [43]. For our purposes, the clarity provided by a simple model to reconstruct RGR and RMF co-variations from sigmoid root and shoot growth trajectories was key. This outweighed the criticism that this is still just another description of data, one that offers no deeper mechanistic insights [43]. Visualising complete co-trajectories of root and shoot growth, rather than disconnected segments of data averaged over long intervals or pertaining to discrete harvests, permitted new insights into how biomass production and allocation are coupled in real time. Our analysis revealed previously uncharacterised, unpredictable and idiosyncratic dynamics of allocation and whole-plant growth. 

A further benefit of using simple phenomenological models with biologically meaningful parameters is that, unlike polynomial functions, they can be used easily to answer “what if?” questions. For example, the species we used in this experiment are common in many grazed habitats in the British Isles, so it would be useful to know how their growth and allocation trajectories might respond to defoliation. This could be done by assuming the almost complete removal of shoot biomass at a certain time (*S* → 0) then plotting regrowth trajectories with the same or different *r* and *Y_max_* values as controls (although, in the case of defoliation, Equation (1) would need to be modified because regrowth, fuelled by the consumption of stored assimilates, tends to follow monomolecular rather than logistic trajectories [17,18,43]).

Are the extrapolated trajectories plotted in Figure 1 and Figure 2 plausible and useful? They are plausible because they are anchored strongly by experimental data (close agreement between data and fitted models), based on biological inevitabilities (sigmoid growth rather than linear or exponential), and produce realistic root–shoot allometries (α and β values similar to those obtained experimentally). If handled judiciously, such extrapolations are useful, first, because they can provide “a quantitative framework of reference” [10] with which to interpret limited experimental data more completely. But extrapolations are no substitute for data from which to derive real and complete trajectories. To obtain such data in similar experiments on these species, the extrapolations reported here suggest that extending experiments to at least 100 d is advisable, increasing the number and frequency of harvests accordingly [55].

Second, and more importantly, the extrapolations are useful because *they highlight the kind of important information about plant biomass allocation that has previously been*
*missed*. That information had been missed simply because (1) experiments (including ours) have been too short and (2) data have been analysed without applying models that explicitly reflect the inevitability of sigmoid growth. Longer experiments, extending well into the phase when root and shoot growth are reaching their maxima, obviously are needed. But until those experiments are done, appropriate extrapolations of limited data are the only tools we have available to glimpse what might be happening in later phases of growth. Such experiments will allay understandable concerns that the new insights about biomass allocation we describe here cannot be taken seriously because they are based, for the time being at least, merely on extrapolated trajectories and not on data that cover a longer period of post-germination development than was possible in our experiment.

We cannot emphasize too strongly that the sole purpose of using the extrapolations was to allow biomass allocation during later phases of growth to be opened up for closer critical scrutiny; it was emphatically not to precisely “predict” root and shoot growth during those phases.

What does allometry tell us about ontogenetic allocation? The strengths and limitations of allometry in characterising biomass allocation in plants have been reviewed extensively [4,5,6,7,8,48,55,56]. Allometry is informative in ways denied to analyses using only time as the independent variable. But allometry, relying mainly [35] if not exclusively [7,20,48] on log–linear relationships among variables, obscures the rich temporal dynamics that roots and shoots can exhibit. Allometric statistics alone (α and β in Appendix A) give little of the flavour of the actual growth trajectories from which they were derived (Figure 1 and Figure 2). That said, allometry is preferable to simple and possibly misleading comparisons of root and shoot biomasses at only a single time or averaged over a long interval. 

Are hydroponically grown plants ecologically relevant? Justification for the use of hydroponics was given in the Introduction, but plants grown hydroponically are unlike their field-grown counterparts in many ways. Testing some of the predictions of this experiment in the field, or at least in soil, using a similar experimental design would be valuable. Few such experiments have been done and are more difficult to do in the field than laboratory. It is, however, possible to subject intact vegetation to perturbations such as warming, irrigation, nutrient additions, shade and elevated atmospheric carbon dioxide concentration, for example. It is important to see the full temporal responses of ecosystems (including biomass allocation) to such factors as opposed to assessing treatment differences at a single arbitrary time after perturbation [12]. Our analysis supports that argument strongly. 

### 4.6. Sigmoid Growth Can Constrain Ontogenetic Allocation—and Its Interpretation

Because a root–shoot allocation response depends on a change in root or shoot growth rates (see Introduction), the possible extent of any such response depends where the plant is along its sigmoid trajectory. In terms of Equation (1), when root and shoots approach their ultimate biomasses (*Y* → *Y_max_*), their absolute growth rates approach zero, i.e., *rY*(1 − (*Y*/*Y_max_*)) → 0, preventing any further influence of growth rate on biomass. Root–shoot allocation is then determined only by *Y_max_*, not growth rate (an exception would be if *Y_max_* increased during ontogeny, if, for example, a large gap formed in an otherwise closed canopy, or nutrient availability increased suddenly, the new conditions then stimulating subsequent growth).

To illustrate this, RMF in *Holcus* would probably have converged towards 0.12 in the control had the experiment lasted for 80 d or longer (Figure 1v). That value is the quotient of root *Y_max_* (1.10 g) and the sum of root and shoot *Y_max_* values (1.10 + 7.93 g); growth rate could not have influenced RMF in *Holcus* plants older than circa 80 d. This is an unavoidable consequence of a sigmoid growth trajectory, one not previously recognised in this context (although implied in Reference [56]). It has important implications for interpreting root–shoot allocation data. 

Measurements of the RMFs of plants, in which the production of new shoot and root material is balanced by senescence or mortality of older structures and have little or no net increase in total biomass, contain different information from RMFs measured on plants capable of some net growth. In the former, RMF reflects mainly the attainable above- and belowground net productivities in that environment, as embodied here in *Y_max_* values, as well as some information about the legacy of previous responses (which will be unknown unless growth has been followed over time); it can say nothing about current responses. Estimates of RMF in a mature forest, for example, will tell you about current above- and belowground net productivities of that forest and perhaps about how the trees might have adjusted their allocation previously in response to climate, competitors or caterpillars, but they cannot tell you how the trees are adjusting their allocation now. In contrast, RMFs of plants in which neither root nor shoot biomass is near its maximum reflect current allocation. However, unless defined as deviations from some control, those RMFs could reflect mainly ontogenetic drift rather than genuine responses to the environment. 

Information about root–shoot allocation derived by following the growth dynamics of individuals (as we did) is not equivalent to that obtained from static interspecific comparisons of “snapshot” measurements of plants differing widely in size [8]. These two approaches might be characterised loosely as collecting ontogenetic or phylogenetic data, respectively, about plant productivity and its allocation. Here, we have described a way to better understand ontogenetic allocation patterns. Phylogenetic information about biomass allocation is also important, not least in helping to predict how vegetation will be influenced by, and will influence, climate [57]. But ontogenetic information and phylogenetic information about root–shoot allocation are distinct and complementary, forming the warp and weft of a fabric fit for the imperial wardrobe. 

## Figures and Tables

**Figure 1 plants-08-00212-f001:**
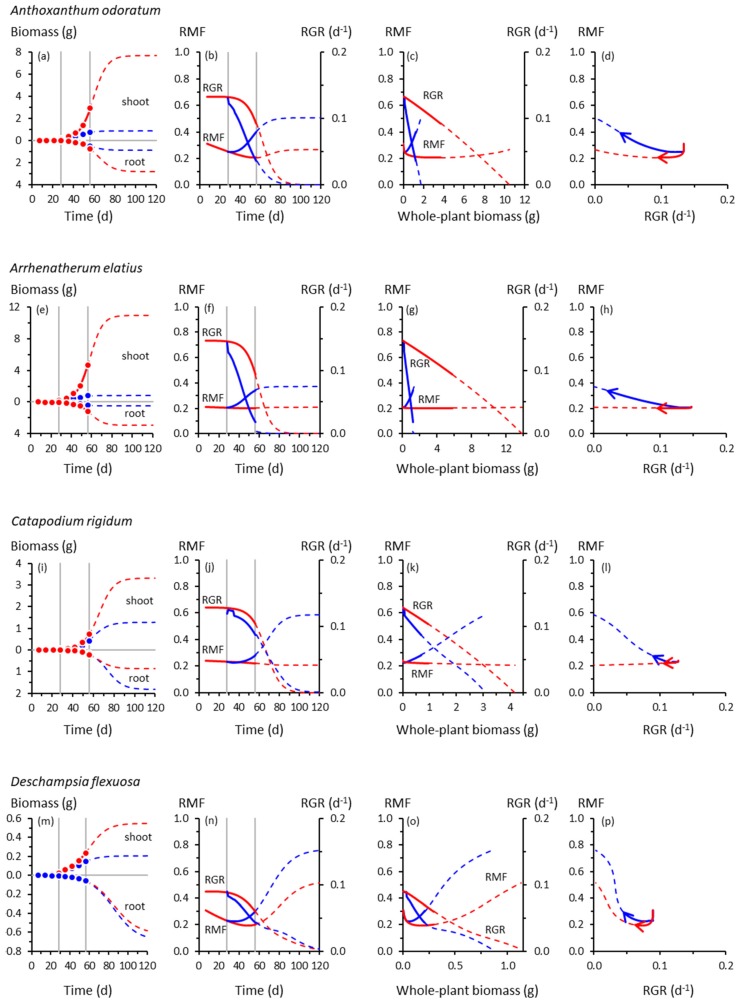
Root and shoot biomass trajectories and corresponding trajectories of root mass fraction (RMF) and whole-plant relative growth rate (RGR) in seven grass species. (**a**) Root and shoot biomass (dry weight) trajectories of *Anthoxanthum odoratum* as functions of time. Symbols are means of experimental data (Appendix A). Red = control, 20 °C day/15 °C night throughout; blue = cooling treatment, 20 °C/15 °C until 28 d, then 10 °C/5 °C from 28 d. Vertical lines indicate the start of cooling at 28 d and the end of the experiment at 56 d. Solid curves show Equation (1) fitted to those data (see Appendix A for fitted parameters and *R*^2^ values). Broken curves are extrapolations of Equation (1) beyond 56 d. (**b**) Trajectories of instantaneous RMF (Equation (2)) and RGR (Equation (3)) of *Anthoxanthum odoratum* as functions of time, derived from the curves plotted in (**a**). Broken curves are extrapolations of Equations (2) and (3) beyond 56 d. (**c**) As for (**b**), but plotted as functions of whole-plant (i.e., root + shoot) biomass. (**d**) Co-variation of instantaneous RMF and RGR for *Anthoxanthum odoratum*; arrows show the direction of temporal progression. Symbols are omitted from panels (**b**)–(**d**) for clarity. Trajectories for other grass species are plotted similarly: (**e**)–(**h**) *Arrhenatherum elatius*; (**i**)–(**l**) *Catapodium rigidum*; (**m**)–(**p**) *Deschampsia flexuosa*; (**q**)–(**t**) *Festuca ovina*; (**u**)–(**x**) *Holcus lanatus*; (**y**)–(**bb**) *Poa annua*. Note the different scales on the biomass axes.

**Figure 2 plants-08-00212-f002:**
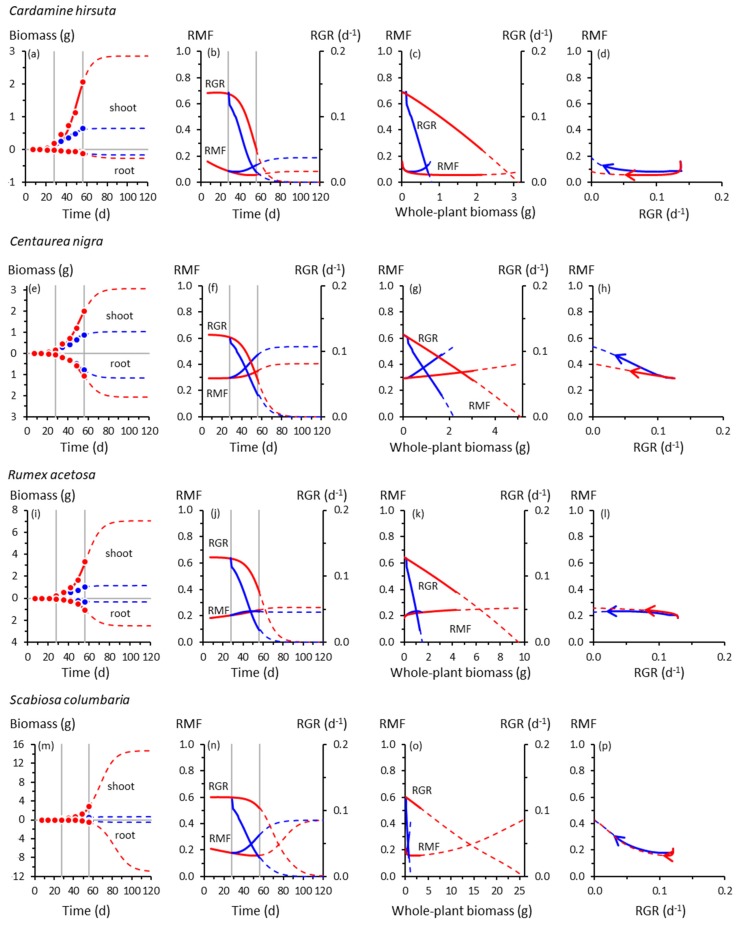
As for Figure 1, for forb species: (**a**)–(**b**) *Cardamine hirsuta*; (**e**)–(**h**) *Centaurea nigra*; (**i**)–(**l**) *Rumex acetosa*; (**m**)–(**p**) *Scabiosa columbaria*.

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
