# Peer review of "Clothing the Emperor: Dynamic Root–Shoot Allocation Trajectories in Relation to Whole-Plant Growth Rate and in Response to Temperature"

_plants, 2019, doi:10.3390/plants8070212_

Round 1

Reviewer 1 Report

REVIEW PLANTS-539798. Clothing the Emperor: dynamic root-shoot allocation trajectories in relation to whole-plant growth rate and in response to temperature

20/6/2019.

This study aimed to compare the root-shoot biomass allocation in individual plants over time in relation to species and size and in response to cooling. The information about root:shoot allocation was obtained by monitoring the growth dynamics of individual plants, thus providing a real insight into ontogenic allocation patterns. The results presented go beyond a mere snapshot of the interspecific comparison of differently sized plants at a specific time. The data usually provided in this type of study represent the biomass distribution, rather than the real biomass allocation, as estimated here.

Given the scarcity of reliable data on biomass allocation for whole plants (in the sense of transfer of new biomass increments), I found this study very useful. The manuscript is very well written (real English is evidently present) and provides a sound discussion of the results in the context of what is already known about co-variation between allocation and whole-plant growth rate, compensatory adjustments in root:shoot allocation and differences in the strategy of growth rate among herbaceous species.

My only concern in reviewing the manuscript is the data extrapolation that the authors carried out after fitting the sigmoid difference model (1) (line 146). Looking at the information shown in Figure 1 and in Table S2, it is clear that in these first 56 days the experiment only covered the initial part of the whole sigmoid trajectory, and the extrapolation therefore starts even before the inflection point of the curve. I wonder why the authors did not consider an experimental framework that covered the 120 days they refer to, without the need for data extrapolation. Of course, a rapid shift from vegetative growth to flowering may have occurred, but this would have been within the observational part of the study.

With the intrinsic early growth information available, it appears very difficult to derive the asymptotic sizes Ymax for shoots and roots. Moreover, both r and Ymax are directly involved in the relationship used to predict plant growth, as the absolute growth is obtained from the ratio between the actual size and the maximum size. The values of determination coefficients tell us very little about this problem, as although they may be close to 1, the estimated Ymax value would be very unreliable. Confidence intervals for parameters r and Ymax should therefore also have been provided.

Another reason that I am skeptical about the extrapolation procedure is that only 4 observations of shoot&root biomass were made after cooling. If the biomass trajectories are supposed to change after the temperature is lowered, the trends would have to be followed for a longer period to ascertain how they actually change. Otherwise, we would be guessing what the trend is on the basis of only 4 points in the time series.

The authors are probably aware of these limitations, as they give as many as 6 reasons to justify the procedure used (lines 162-175). After reading all of these reasons several times, I found none convincing enough to justify not lengthening the experiment for several months and then using the observational values, which provide real evidence of root and shoot biomass allocation, RMF, RGR as a function of time or whole plant biomass or the co-variation of RGR and RMF. For example: how can we be sure that the biomass trajectories for cooled plants do not converge towards those of the control?

Having dealt with root biomass just in a few studies myself and being all of them mere snapshot of tree biomass distribution, it is very difficult for me daring to provide a decision about this manuscript. In my opinion, the relevance of the topic and the possibility of providing information about the “Emperor’s real clothes” are worthy of a longer (and hopefully complete) time series experiment for these herbaceous species.

Author Response

Our responses to the comments below are in red.

This study aimed to compare the root-shoot biomass allocation in individual plants over time in relation to species and size and in response to cooling. The information about root:shoot allocation was obtained by monitoring the growth dynamics of individual plants, thus providing a real insight into ontogenic allocation patterns. The results presented go beyond a mere snapshot of the interspecific comparison of differently sized plants at a specific time. The data usually provided in this type of study represent the biomass distribution, rather than the real biomass allocation, as estimated here.

Given the scarcity of reliable data on biomass allocation for whole plants (in the sense of transfer of new biomass increments), I found this study very useful. The manuscript is very well written (real English is evidently present) and provides a sound discussion of the results in the context of what is already known about co-variation between allocation and whole-plant growth rate, compensatory adjustments in root:shoot allocation and differences in the strategy of growth rate among herbaceous species.

Thank you for that very positive opinion.

My only concern in reviewing the manuscript is the data extrapolation that the authors carried out after fitting the sigmoid difference model (1) (line 146). Looking at the information shown in Figure 1 and in Table S2, it is clear that in these first 56 days the experiment only covered the initial part of the whole sigmoid trajectory, and the extrapolation therefore starts even before the inflection point of the curve. I wonder why the authors did not consider an experimental framework that covered the 120 days they refer to, without the need for data extrapolation. Of course, a rapid shift from vegetative growth to flowering may have occurred, but this would have been within the observational part of the study.

No plants flowered during the experiment.

With the intrinsic early growth information available, it appears very difficult to derive the asymptotic sizes Ymax for shoots and roots. Moreover, both r and Ymax are directly involved in the relationship used to predict plant growth, as the absolute growth is obtained from the ratio between the actual size and the maximum size. The values of determination coefficients tell us very little about this problem, as although they may be close to 1, the estimated Ymax value would be very unreliable. Confidence intervals for parameters r and Ymax should therefore also have been provided.

Yes, that would have been ideal. But, as we state in Section 2.2, the only data available from the experiment (which was done in the late 1970s), were the mean values. The raw data have been lost, regrettably. Without the data from replicates it is not possible to estimate statistical uncertainties on the fitted values of r or Ymax.

Another reason that I am skeptical about the extrapolation procedure is that only 4 observations of shoot&root biomass were made after cooling. If the biomass trajectories are supposed to change after the temperature is lowered, the trends would have to be followed for a longer period to ascertain how they actually change. Otherwise, we would be guessing what the trend is on the basis of only 4 points in the time series.

The authors are probably aware of these limitations, as they give as many as 6 reasons to justify the procedure used (lines 162-175). After reading all of these reasons several times, I found none convincing enough to justify not lengthening the experiment for several months and then using the observational values, which provide real evidence of root and shoot biomass allocation, RMF, RGR as a function of time or whole plant biomass or the co-variation of RGR and RMF. For example: how can we be sure that the biomass trajectories for cooled plants do not converge towards those of the control?

We cannot be sure of this, obviously. Yes, it would have been wonderful to have lengthened the experiment to 120 days or more to avoid any need for extrapolations (but not the need to apply appropriate growth models, even with their inevitable built-in assumptions). We can’t change the design and duration of an experiment done over 40 years ago. But if our paper and the approach described in it encourages researchers to ensure that in future plant growth experiments are not too short to capture full allocation dynamics, then that alone will justify the approach we took in analysing our limited data and in explaining the implications suggested by the data extrapolations, as imperfect as they are.

Having dealt with root biomass just in a few studies myself and being all of them mere snapshot of tree biomass distribution, it is very difficult for me daring to provide a decision about this manuscript. In my opinion, the relevance of the topic and the possibility of providing information about the “Emperor’s real clothes” are worthy of a longer (and hopefully complete) time series experiment for these herbaceous species.

We agree 100% with this view that ‘the relevance of the topic and the possibility of providing information about the “Emperor’s real clothes” are worthy of a longer (and hopefully complete) time series experiment for these herbaceous species.’ Neither we nor anyone else as far as we know have such a complete data set. Ours, as limited as it is in some ways, is the best that is available for this kind of analysis.

Extrapolating models beyond the data is normally frowned upon, as we openly acknowledge and justify in this context (line 162 etc.). But that is apparently still insufficiently convincing, despite the reviewer’s highly complimentary words about the novelty and importance of our work at the start of the review. In another attempt to clarify our intentions, we have added the following text to section 4.5 (the emphasis in the first sentence is deliberate):

Second, and more importantly, the extrapolations are useful because they highlight the kind of important information about plant biomass allocation that has previously been missed. That information had been missed simply because (1) experiments (including ours) have been too short, and (2) data have been analysed without applying models that explicitly reflect the inevitability of sigmoid growth. Longer experiments, extending well into the phase when root and shoot growth are reaching their maxima, obviously are needed. But until those experiments are done, appropriate extrapolations of limited data are the only tools we have available to glimpse what might be happening in later phases of growth. Such experiments will allay understandable concerns that the new insights about biomass allocation we describe here cannot be taken seriously because they are based, for the time being at least, merely on extrapolated trajectories and not on data that cover a longer period of post-germination development than was possible in our experiment.

We cannot emphasize too strongly that the sole purpose of using the extrapolations was to allow biomass allocation during later phases of growth to be opened up for closer critical scrutiny; it was emphatically not to somehow ‘predict’ root and shoot growth during those phases. 

In the absence of specific and detailed editorial guidance, we feel this is the strongest and most unequivocal response we can make to the above criticism.

Reviewer 2 Report

This is an interesting study on root-shoot biomass allocation. 

I suggest a  few minor changes in the introduction:

L36-37:  “but equally…things”, rephrase this part of the sentence

L64-71: this part does not seem to belong to an introduction.

L76-80: environmental biotic and abiotic stress factors will also affect growth, not only resource depletion.

Author Response

Our responses to the comments below are in red.

This is an interesting study on root-shoot biomass allocation. 

Thank you.

I suggest a  few minor changes in the introduction:

L36-37:  “but equally…things”, rephrase this part of the sentence

Rephrase it how and for what reason? The meaning of the sentence is clear and grammatically correct as it stands.

L64-71: this part does not seem to belong to an introduction.

This paragraph summarises the experimental approach we used, and it is followed by a complementary explanation of the rationale for how we analysed data obtained using that approach: the latter would make little sense without the former information. These parts of the Introduction together explain to the reader what we did and why we did it. For that reason, we have retained the text.

L76-80: environmental biotic and abiotic stress factors will also affect growth, not only resource depletion.

The list is not meant to be exhaustive (hence the use of ‘including’ in line 77), but we have added ‘and numerous environmental and biotic constraints’ to line 79-80 to make the point.

Round 2

Reviewer 1 Report

The reply to reviewers sound convincing to me. I think that, with the additional clarifications, the paper provides very useful information on real biomass allocation to root and shoot in herbaceous plants. I would definitevely recommend publishing the ms.